# The SQEIRP Mathematical Model for the COVID-19 Epidemic in Thailand

Sowwanee Jitsinchayakul, Usa Wannasingha Humphries * and Amir Khan

Department of Mathematics, Faculty of Science, King Mongkut's University of Technology Thonburi, 126, Pracha-Uthit Road, Bang Mod, Thung Khru, Bangkok 10140, Thailand
* Correspondence: usa.wan@kmutt.ac.th

**Abstract:** The spread of COVID-19 started in late December 2019 and is still ongoing. Many countries around the world have faced an outbreak of COVID-19, including Thailand, which must keep an eye on the spread and find a way to deal with this extreme outbreak. Of course, we are unable to determine the number of people who will contract this disease in the future. Therefore, if there is a tool that helps to predict the outbreak and the number of people infected, it will be able to find preventive measures in time. This paper aims to develop a mathematical model suitable for the lifestyle of the Thai population facing the COVID-19 situation. It has been established that after close contact with an infected person, a group of individuals will be quarantined and non-quarantined. If they contract COVID-19, they will enter the incubation period of the infection. The incubation period is divided into the quarantine class and the exposed class. Afterwards, both classes will move to the hospitalized infected class and the infected class, wherein the infected class is able to spread the disease to the surrounding environment. This study describes both classes in the SQEIRP model based on the population segmentation that was previously discussed. After that, the positive and bounded solutions of the model are examined, and we consider the equilibrium point, as well as the global stability of the disease-free point according to the Castillo-Chavez method. The SQEIRP model is then numerically analyzed using MATLAB software version R2022a. The cumulative percentage of hospitalized and non-hospitalized infections after 7 days after the commencement of the infection was determined to be 11 and 34 percent of the entire population, respectively. The Next-Generation Matrix approach was used to calculate the Basic Reproduction Numbers ($R_0$). The SQEIRP model's $R_0$ was 3.78, indicating that one infected individual can result in approximately three additional infections. The results of this SQEIRP model provide a preliminary guide to identifying trends in population dynamics in each class.

**Keywords:** coronavirus disease 2019 (COVID-19); SQEIRP epidemic model; reproduction numbers ($R_0$)

**MSC:** 03C98





## 1. Introduction

Humans are social animals. The existence of human society may lead to the emergence of contagions [1]. Some contagions can cause epidemics that spread rapidly among many people in the community at the same time, such as Spanish Flu, SARS, H7N9, Ebola, Zika, MERS, and others [2].

In December 2019, the coronavirus disease 2019 (COVID-19) was discovered. COVID-19 is caused by the severe acute respiratory syndrome coronavirus 2 (SARS-CoV-2), previously provisionally known as the 2019 novel coronavirus (2019-nCoV) [3–6]. Subsequently, this virus spread rapidly from human to human and spread to many countries around the world, causing the World Health Organization (WHO) to declare a global health emergency in late January 2020 [7–9] due to the rapid increase in the number of infected people. Consequently, the number of medical personnel, as well as the amount of medication and medical

equipment, were insufficient to cope with the increasing number of infected people [10,11]. Many countries are interested in the spread of this virus and are urgently trying to find ways to support the growing number of patients to prevent increased mortality. According to the WHO report on 8 December 2022, the total number of cases worldwide has reached 642,924,560 and there have been over 6,625,029 deaths. In Thailand, there were 4,711,528 cases in total, with 33,285 deaths [12]. It is anticipated that the number of people infected and deaths will undoubtedly increase in the future if we are unable to control this virus.

Transmission of SARS-CoV-2 occurs when infected people cough and sneeze. After that, contamination of the virus with these aerosols can cause infection in the air or environment around the infected individual [13–15]. For this reason, if an infected person has contact with or lives with others, the infection will easily spread to others. Thus, there have been many SARS-CoV-2 infections. The diagnosis of COVID-19 requires either laboratory testing or the use of specialized detection equipment [6,16–18]. Typical symptoms of people who contract COVID-19 include fever, coughing, sneezing, shortness of breath, sore throat, loss of sense of smell or taste, as well as complications such as pneumonia. Moreover, the virus can affect the respiratory system of the infected person as well [19]. The incubation period after this virus enters a patient's body is approximately 2 to 14 days after infection, but there is an incubation period of 5 to 6 days in most cases [9,20]. As I mentioned earlier, the incubation period of COVID-19 and specialized diagnostic tests is quite long. Therefore, it is difficult to detect the infected person in a timely manner.

The spread of the virus not only affects the medical field. Other areas, such as transportation, industries, public health, consumption, agriculture, pandemic vulnerability index (PVI), and other fields are also affected [21]. Many countries around the world have been affected in this way. The outbreak illustrates the urgent need for public health response and preparedness to help mitigate the impact of COVID-19. In each country, it is imperative to assess the situation and find appropriate ways to control the outbreak in a timely manner. However, it is difficult to predict future situations in order to mitigate their potential impact. As such, if there is a tool that helps to assess the situation in advance, it will be possible to find ways to prevent the outbreak in a timely manner.

For this reason, one of the interesting tools for assessing the epidemic situation is a mathematical model. Mathematical models are applications of mathematics to describe natural problems such as biology, chemistry, environmental engineering, environmental ecology, physics, social sciences, statistics, wildlife management statistics, etc. In this epidemic situation, mathematical models are tools for predicting future epidemics. The result will guide the development of strategies for the prevention and elimination of epidemics [22,23]. For instance, what is the trend of increasing the number of infected people, so that each country finds a suitable way to prevent the number of infected people, leading to a decrease in the number of deaths? Therefore, the model is valuable for assessing the epidemic situation, providing guidelines for finding appropriate measures to control the epidemic. Mathematical models are abstract structures that use mathematical concepts to describe and develop behavioral systems [6,22]. The mathematical tools convert real situations into problems that mimic real situations, which solve problems mathematically and then interpret them to predict events in real situations [24–26].

Therefore, this paper aims to develop a mathematical model for the COVID-19 epidemic in Thailand. This model is based on the SEIR epidemic model, together with the quarantined population and the pathogens that are transmitted from the infected population to the environment. The model is illustrated in the Materials and Methods section.

## 2. Materials and Methods

The aim of this paper is to develop mathematical models for the COVID-19 epidemic in Thailand. The methodology is as follows. First, we study the epidemic of COVID-19 and the epidemic model. Next, we develop a mathematical model for the COVID-19 epidemic. After that, we perform a stability analysis to verify the stability of the model and find the

basic reproduction number ($R_0$) to determine the degree of infection. Finally, we consider finding solutions from the model to assess the COVID-19 outbreak.

### 2.1. Model Development for COVID-19

Several models have been used to study the COVID-19 epidemic such as SIR, SIRS, SEIR, and others. This research studies the spread of COVID-19 by dividing the human population into six main groups: susceptible class ($S$), quarantine class ($Q$), exposed class ($E$), hospitalized infected class ($I_h$), infected class ($I$), and recovered class ($R$), in conjunction with pathogens ($P$) in the environment spread by an infected class as shown in Figure 1. The model name is SQEIRP, and the model was presented in Equation (1) and Equations (3)–(8). The idea for this model comes from Samuel Mwalili and Ebrahem A. Algehyne, who studied the spread of COVID-19 using the SEIRP model in 2020 and SQIR model in 2021, respectively.

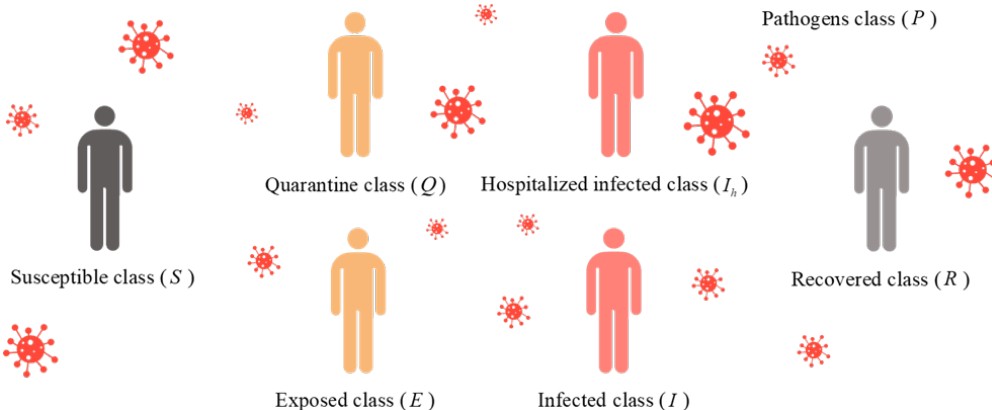

**Figure 1.** The population class of SQEIRP model used to determine the COVID-19 epidemic.

In the first section, consider the susceptible class as representing a healthy or disease-free population. The susceptible class is transferred to the incubation period after exposure to an infected population or to pathogens transmitted from an infected population to the environment. Another inevitable factor is that population dynamics in the susceptible class depend on the natural birth and death of the population. The dynamic of the susceptible class is shown in Equation (1).

$$\frac{dS}{dt} = A + mR - \frac{\sigma \alpha SP}{1 + \omega_1 P} - \frac{\sigma \beta S(I + I_h)}{1 + \omega_2(I + I_h)} - \frac{(1 - \sigma)\alpha SP}{1 + \omega_1 P} - \frac{(1 - \sigma)\beta S(I + I_h)}{1 + \omega_2(I + I_h)} - dS \quad (1)$$

$$\frac{dS}{dt} = A + mR - \frac{\alpha SP}{1 + \omega_1 P} - \frac{\beta S(I + I_h)}{1 + \omega_2(I + I_h)} - dS \quad (2)$$

Second, consider the incubation period population. The population during the incubation period is divided into two classes: quarantine and exposed classes. Both classes represent the populations in quarantine and not in quarantine during the incubation period, respectively. After that, the quarantine and exposed classes are transformed into the hospitalized infected and infected classes, respectively, but if symptoms are severe, some exposure classes may require hospitalization. In addition, consider that the population of quarantine and exposed classes has decreased due to natural mortality. The dynamic of the quarantine and exposed class are shown in Equations (3) and (4), respectively.

$$\frac{dQ}{dt} = \frac{\sigma \alpha SP}{1 + \omega_1 P} + \frac{\sigma \beta S(I + I_h)}{1 + \omega_2(I + I_h)} - \gamma_1 Q - dQ \quad (3)$$

$$\frac{dE}{dt} = \frac{(1 - \sigma)\alpha SP}{1 + \omega_1 P} + \frac{(1 - \sigma)\beta S(I + I_h)}{1 + \omega_2(I + I_h)} - \theta \gamma_2 E - (1 - \theta)\gamma_2 E - dE \quad (4)$$

Third, consider the dynamics of the infected population, comprising the hospitalized infected and the infected classes. The increase in both classes as a result of the incubation period converts to the infected class, and then they are converted to the recovered class. Population dynamics in the hospitalized infected and infected classes are also associated with natural mortality and mortality caused by COVID-19. In addition, the infected class can also spread the pathogens to the environment. The dynamic of the hospitalized infected and the infected classes are shown in Equations (5) and (6), respectively.

$$\frac{dI_h}{dt} = \gamma_1 Q + (1-\theta)\gamma_2 E - \eta_1 I_h - (d+\delta)I_h \tag{5}$$

$$\frac{dI}{dt} = \theta\gamma_2 E - \eta_2 I - (d+\delta)I \tag{6}$$

Next, the part of the recovered class depends on the recovery of the population in the infected class. In this population, if there is a loss of immunity, it becomes the susceptible class. After recovering from COVID-19, permanent immunity is not possible. The dynamic of the infected class is shown in Equation (7).

$$\frac{dR}{dt} = \eta_1 I_h + \eta_2 I - mR - dR \tag{7}$$

Finally, the part of the pathogens class determines the increase in pathogens in the environment caused by coughing or sneezing of the infected class. The pathogens also have a natural death rate. The dynamic of the pathogens class is shown in Equation (8).

$$\frac{dP}{dt} = \mu I - d_p P \tag{8}$$

Here, this research considers the total human population as $N = S + Q + E + I_h + I + R$ and the number of pathogens present in the environment as $P$. The initial conditions are as follows: $S(0) > 0$, $Q(0) > 0$, $E(0) > 0$, $I_h(0) > 0$, $I(0) > 0$, $R(0) > 0$ and $P(0) > 0$. An overview of the population for the spread of COVID-19 is shown in Figure 2.

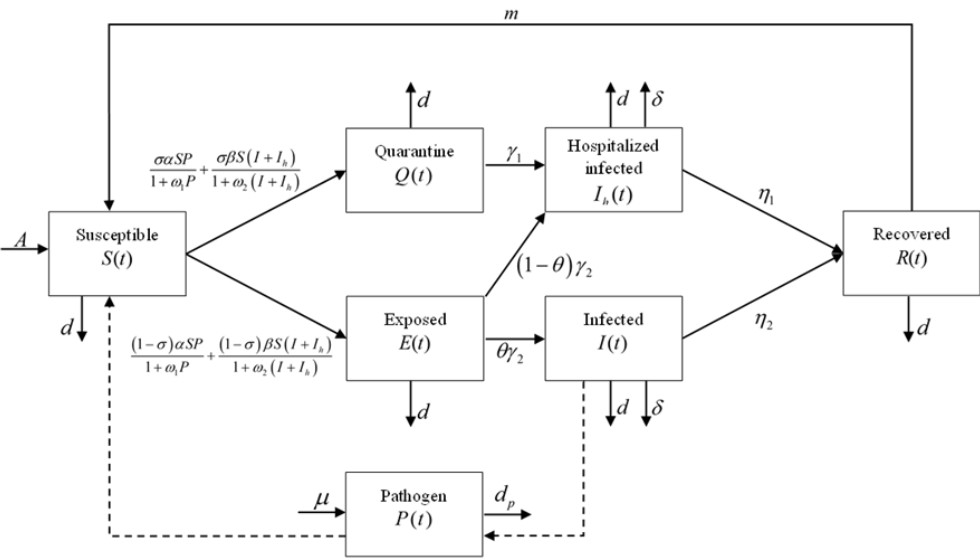

**Figure 2.** The SQEIRP mathematical model for the COVID-19 epidemic.

The parameters and parameter descriptions of the SQEIRP model are shown in Table 1.

**Table 1.** The value of the parameter of the SQEIRP model.

| Parameter Description | Symbol | Value | Source |
|---|---|---|---|
| Birth rate of the human population | $A$ | 0.000028 days$^{-1}$ | [27] |
| Natural human death rate | $d$ | 0.000021 days$^{-1}$ | [27] |
| Probability of the population of the susceptible class that has been quarantined | $\sigma$ | 0.0397 | [28] |
| Probability of the exposed population changing to the infected population | $\theta$ | 0.8 | [28] |
| Rate of transmission from susceptible class to exposed class due to the pathogens | $\alpha$ | 0.00414 days$^{-1}$ | [23] |
| Rate of transmission from susceptible class to exposed class due to the infected | $\beta$ | 0.333333 days$^{-1}$ | [Assume] |
| Proportion of interaction with an infected environment | $\omega_1$ | 0.1 | [23] |
| Proportion of interaction with an infected individual | $\omega_2$ | 0.1 | [23] |
| Rate of transmission from quarantine class to hospitalized infected class | $\gamma_1$ | 1/5.2 days$^{-1}$ | [28] |
| Rate of transmission from exposed class to infected class | $\gamma_2$ | 1/5.2 days$^{-1}$ | [28] |
| Rate of recovery of the hospitalized infected population | $\eta_1$ | 1/10 days$^{-1}$ | [28] |
| Rate of recovery of the infected population | $\eta_2$ | 1/8 days$^{-1}$ | [28] |
| Death rate due to the coronavirus | $\delta$ | 0.00011 days$^{-1}$ | [28] |
| Rate of virus spread to environment by infected class | $\mu$ | 0.1 days$^{-1}$ | [23] |
| Natural death rate of pathogens in the environment | $d_p$ | 0.1724 days$^{-1}$ | [23] |
| Rate of recovered population transferring into a susceptible class | $m$ | 1/90 days$^{-1}$ | [Assume] |

### 2.2. The Preconditions and Boundedness of the SQEIRP Model Solutions

The consideration of positive solutions and the bounded solutions of SQEIRP are considered in Theorems 1 and 2, respectively.

**Theorem 1.** *The solutions of the mathematical model SQEIRP are positive for all $t > 0$.*

**Proof of Theorem 1.** Let $S(0) = S_0 > 0$, $Q(0) = Q_0 > 0$, $E(0) = E_0 > 0$, $I_h(0) = I_{h_0} > 0$, $I(0) = I_0 > 0$, $R(0) = R_0 > 0$ and $P(0) = P_0 > 0$. Considering Equation (2), we get

$$\frac{dS}{dt} \geq -\frac{\alpha S P}{1 + \omega_1 P} - \frac{\beta S(I + I_h)}{1 + \omega_2(I + I_h)} - dS,$$

$$\frac{dS}{dt} \geq \left( -\frac{\alpha P}{1 + \omega_1 P} - \frac{\beta(I + I_h)}{1 + \omega_2(I + I_h)} - d \right) S,$$

$$\frac{1}{S} dS \geq \left( -\frac{\alpha P}{1 + \omega_1 P} - \frac{\beta(I + I_h)}{1 + \omega_2(I + I_h)} - d \right) dt,$$

$$\int \frac{1}{S} dS \geq \int \left( -\frac{\alpha P}{1 + \omega_1 P} - \frac{\beta(I + I_h)}{1 + \omega_2(I + I_h)} \right) dt - \int (d) dt, \tag{9}$$

$$\ln |S| \geq -\left( \frac{\alpha P}{1 + \omega_1 P} + \frac{\beta(I + I_h)}{1 + \omega_2(I + I_h)} \right) t - dt + C,$$

$$S(t) \geq e^{-\left( \frac{\alpha P}{1 + \omega_1 P} + \frac{\beta(I + I_h)}{1 + \omega_2(I + I_h)} \right) t - dt + C},$$

$$S(t) \geq S_0 e^{-\left(\frac{\alpha P}{1+\omega_1 P} + \frac{\beta(I+I_h)}{1+\omega_2(I+I_h)}\right)t - dt} > 0.$$

As for $Q(t)$, $E(t)$, $I_h(t)$, $I(t)$, $R(t)$ and $P(t)$, considering Equations (3)–(8), respectively, results in the following

$$Q(t) \geq Q_0 e^{-(\gamma_1 + d)t} > 0, \tag{10}$$

$$E(t) \geq E_0 e^{-(\gamma_2 + d)t} > 0, \tag{11}$$

$$I_h(t) \geq I_{h_0} e^{-(\eta_1 + d + \delta)t} > 0, \tag{12}$$

$$I(t) \geq I_0 e^{-(\eta_2 + d + \delta)t} > 0, \tag{13}$$

$$R(t) \geq R_0 e^{-(m+d)t} > 0, \tag{14}$$

and

$$P(t) \geq P_0 e^{-d_p t} > 0. \tag{15}$$

Then, $S(t)$, $Q(t)$, $E(t)$, $I_h(t)$, $I(t)$, $R(t)$ and $P(t)$ of the SQEIRP mathematical model are non-negative for all $t > 0$.  $\square$

**Theorem 2.** *The solution for the SQEIRP model is within the boundary*

$$\Omega = \left\{ (S, Q, E, I_h, I, R, P) \in \mathbb{R}^7 : 0 \leq N(t) \leq \frac{A}{d}, 0 \leq P(t) \leq \frac{\mu A}{d d_p} \right\}.$$

**Proof of Theorem 2.** For the human population, let $N(t) = S(t) + Q(t) + E(t) + I_h(t) + I(t) + R(t)$ and $N(0) = N_0$.

Consider
$$\frac{dN}{dt} = \frac{dS}{dt} + \frac{dQ}{dt} + \frac{dE}{dt} + \frac{dI_h}{dt} + \frac{dI}{dt} + \frac{dR}{dt},$$

$$\frac{dN}{dt} = A - dS - dQ - dE - (d + \delta)I_h - (d + \delta)I - dR,$$

$$\frac{dN}{dt} = A - d(S + Q + E + I_h + I + R) - \delta(I_h + I), \tag{16}$$

$$\frac{dN}{dt} = A - dN - \delta(I_h + I); N = S + Q + E + I_h + I + R,$$

$$\frac{dN}{dt} \leq A - dN.$$

Therefore,
$$N(t) \leq \frac{A}{d}(1 - e^{-dt}) + N_0 e^{-dt},$$

and get,
$$\limsup_{t \to \infty} N(t) \leq \limsup_{t \to \infty} \left( \frac{A}{d}(1 - e^{-dt}) + N_0 e^{-dt} \right) = \frac{A}{d}.$$

For the pathogen population P(t), let $P(0) = P_0$.

Consider
$$\frac{dP}{dt} = \mu I - d_p P,$$

$$\frac{dP}{dt} \leq \mu \frac{A}{d} - d_p P.$$

Therefore,
$$P(t) \leq \frac{\mu A}{d_p d}(1 - e^{-d_p t}) + P_0 e^{-d_p t},$$

and get,
$$\limsup_{t \to \infty} P(t) \leq \limsup_{t \to \infty} \left( \frac{\mu A}{d_p d}(1 - e^{-d_p t}) + P_0 e^{-d_p t} \right) = \frac{\mu A}{d_p d}.$$

$\square$

### 3. Stability Analysis of Development for COVID-19

This research presents the SQEIRP model for the COVID-19 epidemic, as shown in Equation (1) and Equation (3) through (8). This research examines the behavior of this model using stability analysis. First, we find the equilibrium point of the system of equations of the developed COVID-19 model. Then, the stability by the roots of the characteristic equation is determined by considering $|J - \lambda I| = 0$.

#### 3.1. Equilibrium Point of the SQEIRP Model

The equilibrium point of the SQEIRP model for the COVID-19 epidemic can be found by assigning the derivative of the model equal to 0.

#### 3.1.1. Disease-Free Equilibrium Point of the SQEIRP Model

The disease-free point of the SQEIRP model is

$$X^0 = (S^*, Q^*, E^*, I_h{}^*, I^*, R^*, P^*) = \left(\frac{A}{d}, 0, 0, 0, 0, 0, 0\right).$$

#### 3.1.2. Epidemic Equilibrium Point of the SQEIRP Model

The epidemic equilibrium point of the SQEIRP model is

$$X^* = (S^*, Q^*, E^*, I_h{}^*, I^*, R^*, P^*),$$

where

$$S^* = \frac{A}{d} + \frac{m}{d}R^* - \frac{\gamma_1 + d}{d}Q^* - \frac{\gamma_2 + d}{d}E^*, Q^* = \frac{\eta_1 + d + \delta}{\gamma_1}I_h{}^* - \frac{(1-\theta)\gamma_2}{\gamma_1}E^*, E^* = \frac{\eta_2 + d + \delta}{\theta\gamma_2}I^*,$$

$$I_h{}^* = \frac{A - dN^*}{\delta} - I^*, I^* = \frac{-A_2 + \sqrt{A_2{}^2 - 4A_1A_3}}{2A_1}, R^* = \frac{\eta_1 I_h{}^* + \eta_2 I^*}{m + d}, P^* = \frac{\mu I^*}{d_p},$$

and

$$A_1 = ((1-\sigma)\alpha\mu + y_1\omega_1\mu)(-y_2 + y_3) - y_4\omega_1\mu,$$

$$A_2 = ((1-\sigma)\alpha\mu + y_1\omega_1\mu)\left(A + y_2\left(\frac{A - dN^*}{\delta}\right)\right) - y_1d_p(-y_2 + y_3) - y_4d_p,$$

$$A_3 = y_1d_p\left(A + y_2\left(\frac{A - dN^*}{\delta}\right)\right),$$

with

$$y_1 = \frac{(1-\sigma)\beta(A - dN^*)}{\delta + \omega_2(A - dN^*)}, y_2 = \frac{m\eta_1}{m + d} - \frac{(\gamma_1 + d)(\eta_1 + d + \delta)}{\gamma_1},$$

$$y_3 = \frac{m\eta_2}{m + d} + \frac{\eta_2 + d + \delta}{\theta\gamma_2}\left(\frac{(1-\theta)\gamma_2(\gamma_1 + d)}{\gamma_1} - (\gamma_2 + d)\right), y_4 = \frac{d(\gamma_2 + d)(\eta_2 + d + \delta)}{\theta\gamma_2}.$$

#### 3.2. Basic Reproduction Numbers ($R_0$) of the SQEIRP Model

The value of $R_0$ is calculated from the next-generation matrix ($G$) method, as follows. First, consider

$$\frac{dX}{dt} = F_i(X) - V_i(X),$$

where $X = [Q, E, I_h, I, P]^t$, $F_i(X)$ is the matrix of the rate of appearance of new infections in compartment $i$ and $V_i(X)$ is the matrix of rate of other transitions between compartment $i$ and other infected compartments. Thus, we get

$$F_i(X) = \begin{bmatrix} \frac{\sigma\alpha SP}{1+\omega_1 P} + \frac{\sigma\beta S(I+I_h)}{1+\omega_2(I+I_h)} \\ \frac{(1-\sigma)\alpha SP}{1+\omega_1 P} + \frac{(1-\sigma)\beta S(I+I_h)}{1+\omega_2(I+I_h)} \\ 0 \\ 0 \\ 0 \end{bmatrix}, V_i(X) = \begin{bmatrix} \gamma_1 Q + dQ \\ \theta\gamma_2 E + (1-\theta)\gamma_2 E + dE \\ \eta_1 I_h + (d+\delta)I_h - \gamma_1 Q - (1-\theta)\gamma_2 E \\ \eta_2 I + (d+\delta)I - \theta\gamma_2 E \\ -\mu I + d_p P \end{bmatrix}.$$

Next, the Jacobian of $F_i(X)$ at the disease-free equilibrium is

$$F = \begin{bmatrix} 0 & 0 & \frac{\sigma\beta A}{d} & \frac{\sigma\beta A}{d} & \frac{\sigma\alpha A}{d} \\ 0 & 0 & \frac{(1-\sigma)\beta A}{d} & \frac{(1-\sigma)\beta A}{d} & \frac{(1-\sigma)\alpha A}{d} \\ 0 & 0 & 0 & 0 & 0 \\ 0 & 0 & 0 & 0 & 0 \\ 0 & 0 & 0 & 0 & 0 \end{bmatrix}$$

and the Jacobian of $V_i(X)$ at the disease-free equilibrium is

$$V = \begin{bmatrix} \gamma_1 + d & 0 & 0 & 0 & 0 \\ 0 & \gamma_2 + d & 0 & 0 & 0 \\ -\gamma_1 & -(1-\theta)\gamma_2 & \eta_1 + d + \delta & 0 & 0 \\ 0 & -\theta\gamma_2 & 0 & \eta_2 + d + \delta & 0 \\ 0 & 0 & 0 & -\mu & d_p \end{bmatrix}.$$

Then, the next-generation matrix is

$$FV^{-1} = \begin{bmatrix} \frac{\sigma\beta A\gamma_1}{dk_1k_3} & \frac{\sigma\beta A(1-\theta)\gamma_2}{dk_2k_3} + \frac{\sigma\beta A\theta\gamma_2}{dk_2k_4} + \frac{\sigma\alpha A\theta\gamma_2\mu}{dk_2k_4d_p} & \frac{\sigma\beta A}{dk_3} & \frac{\sigma\beta A}{dk_4} + \frac{\sigma\alpha A\mu}{dk_4d_p} & \frac{\sigma\alpha A}{dd_p} \\ \frac{(1-\sigma)\beta A\gamma_1}{dk_1k_3} & \frac{(1-\sigma)\beta A(1-\theta)\gamma_2}{dk_2k_3} + \frac{(1-\sigma)\beta A\theta\gamma_2}{dk_2k_4} + \frac{(1-\sigma)\alpha A\theta\gamma_2\mu}{dk_2k_4d_p} & \frac{(1-\sigma)\beta A}{dk_3} & \frac{(1-\sigma)\beta A}{dk_4} + \frac{(1-\sigma)\alpha A\mu}{dk_4d_p} & \frac{(1-\sigma)\alpha A}{dd_p} \\ 0 & 0 & 0 & 0 & 0 \\ 0 & 0 & 0 & 0 & 0 \\ 0 & 0 & 0 & 0 & 0 \end{bmatrix},$$

with $k_1 = \gamma_1 + d$, $k_2 = \gamma_2 + d$, $k_3 = \eta_1 + d + \delta$ and $k_4 = \eta_2 + d + \delta$.

Therefore, the basic reproductive number given by the spectral radius of the matrix is

$$R_0 = \rho\left(FV^{-1}\right) = \frac{(1-\sigma)\beta A(1-\theta)\gamma_2}{dk_2k_3} + \frac{(1-\sigma)\beta A\theta\gamma_2}{dk_2k_4} + \frac{(1-\sigma)\alpha A\theta\gamma_2\mu}{dk_2k_4d_p} + \frac{\sigma\beta A\gamma_1}{dk_1k_3}.$$

Global Stability Analysis of Disease-free Equilibrium Point of the SQEIRP Model When $R_0 < 1$

Next, we consider the global stability of the disease-free equilibrium point of the SQEIRP model with the method of Castillo-Chavez et al. [29]. First, the model is divided into two parts, as shown in Equations (17) and (18).

$$\frac{dY_1}{dt} = G(Y_1, Y_2) \tag{17}$$

$$\frac{dY_2}{dt} = H(Y_1, Y_2) \tag{18}$$

where $Y_1$ represents the uninfected populations; $Y_1 = (S, R) \in \mathbb{R}^2$, and $Y_2$ represent the infected populations; $Y_2 = (Q, E, I_h, I, P) \in \mathbb{R}^5$.

Next, the global stability of the disease-free equilibrium was determined according to the following conditions, with the disease-free equilibrium $X^0 = \left(Y_1{}^0, 0\right)$.

(i) For $\frac{dY_1}{dt} = G(Y_1, 0)$, $Y_1{}^0$ is globally asymptotically stable.

(ii) $H(Y_1, Y_2) = BY_2 - \hat{H}(Y_1, Y_2)$ where $hatH(Y_1, Y_2) \geqslant 0$ for all $(Y_1, Y_2) \in \Omega$.

**Theorem 3.** *The disease-free equilibrium point $(X_0)$ of the SQEIRP model is globally asymptotically stable.*

**Proof of Theorem 3.** Let $Y_1 = (S, R) \in \mathbb{R}^2$, $Y_2 = (Q, E, I_h, I, P) \in \mathbb{R}^5$ and $X^0 = \left(Y_1{}^0, 0\right)$; $Y_1{}^0 = \left(\frac{A}{d}, 0\right)$, and get

$$\frac{dY_1}{dt} = G(Y_1, Y_2) = \begin{bmatrix} A + mR - \frac{\alpha SP}{1+\omega_1 P} - \frac{\beta S(I+I_h)}{1+\omega_2(I+I_h)} - dS \\ \eta_1 I_h + \eta_2 I - mR - dR \end{bmatrix}, \tag{19}$$

$$\frac{dY_2}{dt} = H(Y_1, Y_2) = \begin{bmatrix} \frac{\sigma\alpha SP}{1+\omega_1 P} + \frac{\sigma\beta S(I+I_h)}{1+\omega_2(I+I_h)} - \gamma_1 Q - dQ \\ \frac{(1-\sigma)\alpha SP}{1+\omega_1 P} + \frac{(1-\sigma)\beta S(I+I_h)}{1+\omega_2(I+I_h)} - \theta\gamma_2 E - dE \\ \gamma_1 Q + (1-\theta)\gamma_2 E - \eta_1 I_h - (d+\delta)I_h \\ \theta\gamma_2 E - \eta_2 I - (d+\delta)I \\ \mu I - d_p P \end{bmatrix}. \tag{20}$$

First, consider $\frac{dY_1}{dt} = G(Y_1, 0)$ by defining $S = S^0$, $R = R^0$ and $G(Y_1, 0) = 0$, and get

$$G(Y_1, 0) = \begin{bmatrix} A + mR - dS \\ -mR - dR \end{bmatrix} = 0. \tag{21}$$

According to Equation (21), then $Y_1 \rightarrow Y_1{}^0$ as $t \rightarrow \infty$. So $Y_1 = Y_1{}^0$ is globally asymptotically stable.

Next, consider $H(Y_1, Y_2) = BY_2 - \hat{H}(Y_1, Y_2)$ by defining

$$B = \begin{bmatrix} -\gamma_1 - d & 0 & \frac{\sigma\beta A}{d} & \frac{\sigma\beta A}{d} & \frac{\sigma\alpha A}{d} \\ 0 & -\gamma_2 - d & \frac{(1-\sigma)\beta A}{d} & \frac{(1-\sigma)\beta A}{d} & \frac{(1-\sigma)\alpha A}{d} \\ \gamma_1 & (1-\theta)\gamma_2 & -\eta_1 - d - \delta & 0 & 0 \\ 0 & \theta\gamma_2 & 0 & -\eta_2 - d - \delta & 0 \\ 0 & 0 & 0 & \mu & -d_p \end{bmatrix},$$

where $B$ is an M-matrix (the off-diagonal elements are non-negative) and

$$\hat{H}(Y_1, Y_2) = \begin{bmatrix} \frac{\sigma\beta A(I_h+I)}{d} + \frac{\sigma\alpha AP}{d} - \frac{\sigma\alpha SP}{1+\omega_1 P} - \frac{\sigma\beta S(I+I_h)}{1+\omega_2(I+I_h)} \\ \frac{(1-\sigma)\beta A(I_h+I)}{d} + \frac{(1-\sigma)\alpha AP}{d} - \frac{(1-\sigma)\alpha SP}{1+\omega_1 P} - \frac{(1-\sigma)\beta S(I+I_h)}{1+\omega_2(I+I_h)} \\ 0 \\ 0 \\ 0 \end{bmatrix}, \tag{22}$$

and get

$$\frac{\sigma\beta A(I_h + I)}{d} + \frac{\sigma\alpha AP}{d} > \frac{\sigma\alpha SP}{1+\omega_1 P} + \frac{\sigma\beta S(I + I_h)}{1+\omega_2(I + I_h)}, \tag{23}$$

and

$$\frac{(1-\sigma)\beta A(I_h + I)}{d} + \frac{(1-\sigma)\alpha AP}{d} > \frac{(1-\sigma)\alpha SP}{1+\omega_1 P} + \frac{(1-\sigma)\beta S(I + I_h)}{1+\omega_2(I + I_h)}. \tag{24}$$

This implies that $\hat{H}(Y_1, Y_2) \geq 0$ for all $(Y_1, Y_2) \in \Omega$.

Therefore, by proving the above two conditions, the disease-free equilibrium point $(X^0)$ of the SQEIRP model is globally asymptotically stable. $\square$

## 4. Results

The development of a mathematical model for the outbreak of COVID-19 in Thailand was achieved by dividing the population into two groups, such as the human population (*N*) and the pathogen population (*P*) transmitted from infected people to the environment. This article considers the model of the COVID-19 epidemic in Thailand, dividing the incubation period and infected population into two lines. The first line represents the quarantined population. This population has contracted the virus. After division, the population in the incubation period is quarantined in hospitals or government-provided residences and eventually infected. The second line represents the mild or asymptomatic population. This population is incubated but not quarantined. Some infected people are not admitted to hospitals or government-provided residences, so these people can spread the disease into the environment. The non-quarantine population will be hospitalized after they present severe symptoms.

The parameter values and the source parameters of this model are shown in Table 1. The natural birth and death rates of the population in Thailand in 2021 are 10.25 per 1000 people and 7.66 per 1000 people, respectively, with $A = \frac{10.25}{356 \times 1000}$ days$^{-1}$ and $d = \frac{7.66}{356 \times 1000}$ days$^{-1}$ [27]. In addition, those returning to a susceptible class can be re-infected, considering the loss of immunity after recovery of approximately 3 months or 90 days [30]. Therefore, the rate of the recovered population converting to the susceptible class is equal to $1/90$ days$^{-1}$. The model's hospital infection estimate used the parameters shown in Table 1 compared with actual data in Thailand in January 2022, where the COVID-19 variant omicron outbreak was detected [31]. Figure 3 shows the model solution based on actual data from 1 January 2022 to 17 January 2022. Based on this time period, the model's solution compared to the actual data shows that the trend of the graph is in the same direction.

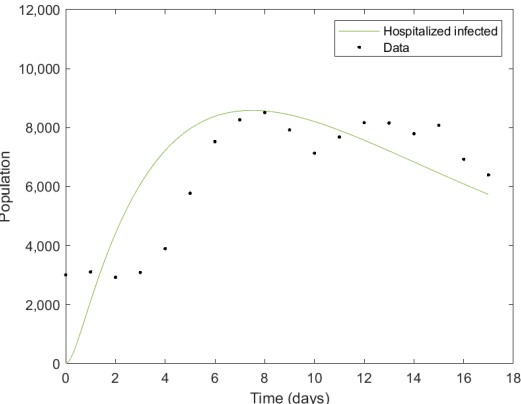

**Figure 3.** The simulation of the hospitalized infected population compared to the actual data from 1 January 2022 to 17 January 2022.

### 4.1. The Numerical Analysis of the SQEIRP Model

The numerical analysis of the SQEIRP model using MATLAB software version R2022a with the initial values of $S(0), Q(0), E(0), I_h(0), I(0), R(0)$, and $P(0)$ are equal to 66,170,000; 12; 6,331; 0; 0; 0; and 500, respectively [23,27,32]. Therefore, the numerical analysis of the SQEIRP model is shown as follows. Figure 4 shows an overview of the dynamics of the human population and the pathogen population from 0–90 days. The dynamics of the human population infected with the virus but during the incubation period and the dynamics of the infected population are shown in Figures 5 and 6, respectively.

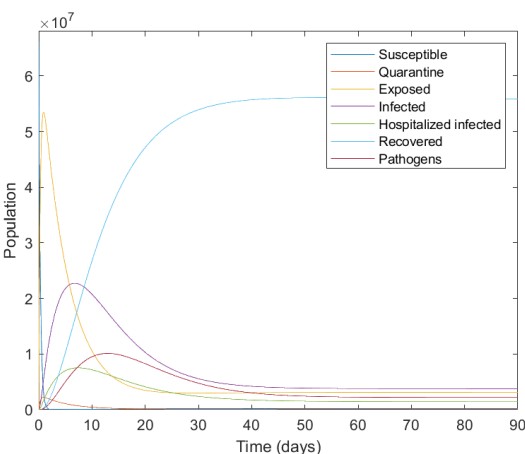

**Figure 4.** The simulation of human and pathogen population from 0 to 90 days.

The dynamic of the susceptible population declines rapidly over a period of 0–3 days, with the initial population set at 66,170,000. This decline in population shifts to populations that are in the incubation period after exposure to the virus. The incubation period population dynamics are divided into two types: quarantine class, and exposed class, as shown in Figure 5a,b respectively. The dynamics of the quarantine and exposed class increase rapidly over the period of 0–1 day. The maximum population of the quarantine class is approximately 2,200,000, which is less than the maximum population of the exposed class of approximately 53,000,000. After day 1, the dynamics of both populations decline rapidly.

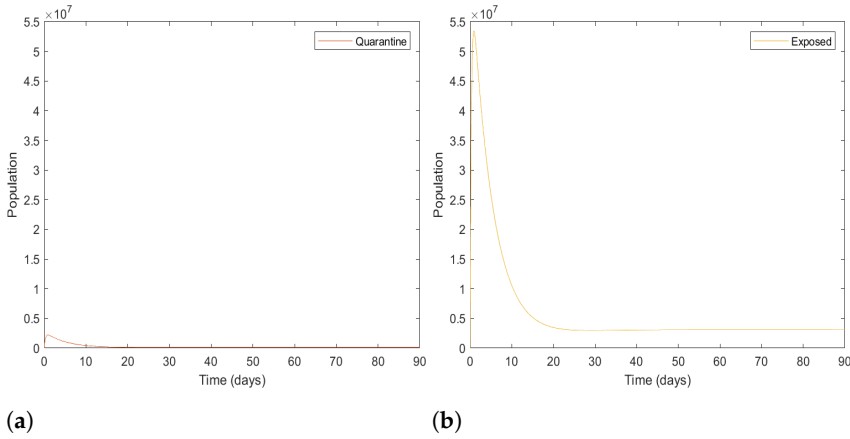

(**a**)                                            (**b**)

**Figure 5.** The simulated of quarantine (**a**) and exposed population (**b**) from 0 to 90 days.

The dynamics of the hospitalized infected and infected population are related to the dynamics of the quarantine and exposed population, respectively. As a result of the incubation period population changes to the infected population, the dynamics of hospitalized infected population and infected population increase rapidly over 0–7 days, with the peak population estimated at 7,500,000 and 22,700,000, respectively. After that, it declines rapidly. The dynamics of the hospitalized infected and infected population are shown in Figure 6a,b respectively.

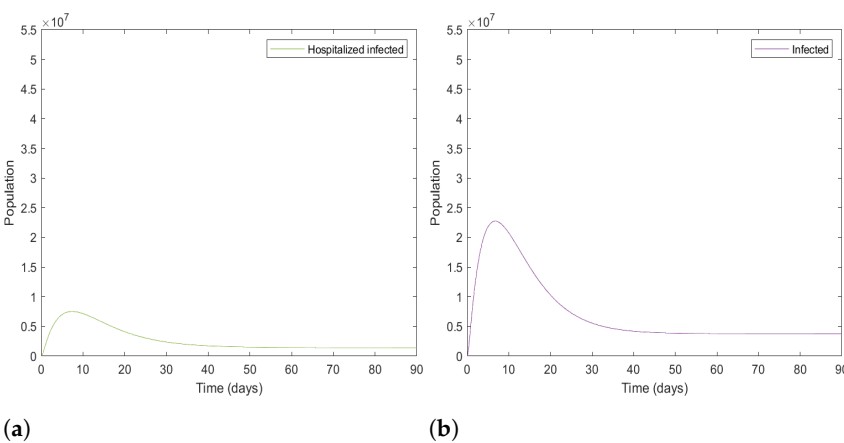

**Figure 6.** The simulated of hospitalized infected (**a**) and infected population (**b**) from 0 to 90 days.

The final part of the human population dynamics is expressed in the recovered population dynamics, increasing gradually over 0–30 days, and after that, it begins to stabilize. Part of the pathogens population in the environment is caused by the infectious population. The initial population of pathogens was set at 500. At 0–12 days, the number of pathogens increases sharply and then decreases. The dynamics of the recovered population and the pathogen population are shown in Figure 4.

*4.2. The Value of Basic Reproduction Numbers ($R_0$)*

The dynamic simulation of each population is shown in Figure 4. This paper considers the basic reproduction numbers ($R_0$) to determine how many infections will originate from one infected person. Where $R_0$ is greater than 1, it means that one infected person can increase the number of infected people to more than one, which could cause an outbreak. Similarly, a value of $R_0$ less than 1 would not result in an outbreak. In this work, the $R_0$ values of the SQEIRP model were considered, together with the parameters assigned to be as relevant as possible for the outbreak in Thailand. The $R_0$ value of the SQEIRP model is 3.78, meaning that one patient can lead to approximately three additional infections.

**5. Discussions**

The concept of pathogens being transmitted to the environment around infected people is supported by Samuel Mwalili's research. The concept of quarantined and non-quarantined population segments is supported by Ebrahem A. Algehyne's research, as well as the COVID-19 situation report provided by the Ministry of Health in Thailand. The parameters used in the model to estimate the epidemic situation in Thailand are presented in Table 1, which refers to the research of Jankhonkhan et al., who studied the spread of COVID-19 in Thailand from 1 November 2020 to 19 August 2021. The parameters also related to the pathogen research by Samuel Mwalili, as well as the report on the COVID-19 epidemic in Thailand. However, this article considers adjusting parameters that correspond to an increase in the number of infected people based on the outbreak of the COVID-19 variant omicron that began to spread in Thailand in early January 2022. Here, we consider the tendency to change the solution of patients admitted to the hospitalization compared to the actual data from 1 January 2022, to 17 January 2022.

The solution of this model only applies to the beginning of the outbreak of the COVID-19 variant omicron, during which time people were very careful. Thus, there was strict prevention of the disease. As a result, the number of infected people was low. Afterward, people reduced epidemic prevention, and new COVID-19 variants occurred; as a result, the number of infected people increased. Therefore, the parameter representing the number of infected people had to be adjusted. The result of this model assigns all populations in the country as a susceptible class. Then, the number of infected people is quite high. Therefore, it is possible to analyze the susceptible class more accurately.

## 6. Conclusions

This article aimed to develop a mathematical model suitable for Thai people's lifestyle during the COVID-19 epidemic in Thailand. The SEIR model was developed alongside the SQEIRP model by adding parts of the quarantine population and the pathogens that spread in the environment. Then, we found parameters suitable for the epidemic situation. The results of the SQEIRP model showed that approximately 7 days after the onset of the infection, the number of hospitalized infected and infected cases was as high as 7,500,000 and 22,700,000 or 11 and 34 percent of the total population, respectively. Moreover, the maximum number of infected people provides a guideline for the government and people to prevent the spread of COVID-19 such that the number of patients does not increase to exceed the estimated number of the model. Therefore, we suggest adjusting parameters according to infected tendencies, including estimating the number of susceptible classes accurately. Thus, the model results will be able to predict the epidemic situation better.

Finally, the researcher expects the SQEIRP model will adapt to the Thai people's lifestyle and the transmission characteristics of COVID-19. This will lead to results used to predict the spread of COVID-19 in Thailand in order to provide guidelines for assessing the transmission situation epidemic and finding ways to prevent the epidemic in a timely manner next time.

**Author Contributions:** writing—original draft preparation, S.J.; writing—review and editing, U.W.H.; writing—review and editing, A.K. All authors have read and agreed to the published version of the manuscript.

**Funding:** This research was funded by Petchara Pra Jom Klao Doctoral Scholarship, King Mongkut's University of Technology Thonburi (KMUTT), Thailand grant number 68/2563.

**Data Availability Statement:** Not applicable.

**Acknowledgments:** We gratefully acknowledge the support from the Petchra Pra Jom Klao Doctoral Scholarship, King Mongkut's University of Technology Thonburi (KMUTT), Thailand.

**Conflicts of Interest:** The authors declare no conflict of interest.

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
