# Peer review of "The SQEIRP Mathematical Model for the COVID-19 Epidemic in Thailand"

_axioms, doi:10.3390/axioms12010075_

Round 1

Reviewer 1 Report

 Manuscript ID : axioms-2119846

 Title: The SQEIRP Mathematical Model for the COVID-19 Epidemic in Thailand

This paper aims to develop a mathematical model suitable for the lifestyle of the Thai population facing the COVID-19 situation. It was found that after close contact with the infected person, they are quarantined and not quarantined. In this paper, the incubation-infected and the infected population of this model are considered into quarantine class, exposed class, hospitalized infected class, and infected class. Including, infection of the population caused by pathogens in the environment. Based on all the population segmentations mentioned, this study defines them in the SQEIRP model. Subsequently, Numerical analysis of the SQEIRP model using MATLAB software version R2022a. It was found that after 7 days after the onset of infection, the cumulative number of hospitalized and non-hospitalized infections was 11 and 34 percent of the total population, respectively. The Basic Reproduction Numbers(R0) were determined using the Next-Generation Matrix method to determine the outbreak severity. The R0 of the SQEIRP model was 3.78 which means one infected person can lead to about three more infections. The results of this SQEIRP model provide a preliminary guide to identifying trends in population dynamics in each class.

Comments

·       This paper aims to develop a mathematical model for COVID-19 based on the SEIR epidemic model in conjunction with quarantined populations and pathogens transmitted to the environment.

·       The idea for this model comes from a study by Samuel Mwalili in 2020, that studied the spread of COVID-19 with the SEIRP model, together with research by Ebrahem A. Algehyne in 2021, that studied the spread of COVID-19 with the SQIR model.

·       This research studies the spread of COVID-19 by dividing the human population into six main groups and not four groups as some previous published papers

·       Developing the model and taking into account all the above considered cases is mathematically right.

·       For the proof of the preconditions and boundedness of the SQEIRP model solutions and the stability analysis, the authors have used the same procedure presented in published paper concerning other models of COVID-19 and the obtained results here can be considered as a general case as shown in numerical results section

The article is a relevant contribution for the journal, so I find it appropriate to be published in Axioms. However, I think that it should be improved before final acceptance by taking into account the following points:

·       Some English statements are difficult to incomprehensible or not suitable and extensive editing of English language and style are required

·       The references list should be updated to include the very recently published/relevant papers

·       The author should explain how the Rate of recovered population go into a susceptible class and some other parameters are assumed

·       Overall, the paper, too much verbosity in explanation should be avoided

·       Validation of the results is very important, especially the authors have adjusted some parameters     

.

Reviewer 2 Report

The issue raised by the authors is indeed an important one, and is interesting from both scientific and practical perspectives.
The literature review refers to the central issue of the paper, it is quite extensive, relevant and thorough. The review will be of interest to other researchers. I would like to mention that the authors have comprehensively studied the literature on the issue published over the last five years. References are correct. The conclusion is consistent with presented arguments and evidence. The results complete previous results on the matter and are supported by references.
However, there is a small comment to the work. The abstract does not convey the content of the paper accurately. The abstract should introduce the reader to the research and hence state the research objective clearly, the methods used, some of the specific findings and some of the conclusions. I would also recommend authors to increase the abstract to 200 words.
In the structure of the paper a Discussion section is absent, though it is rather significant. In this part authors included conclusions. It wasn't worth doing. There is no true discussion. I believe that the section should be included and should contain the comparison of results in the context of previously published literature on the matter, as well as their discussion, which would allow authors to prove their points on the issue under consideration. The discussion does not clearly show the importance of the study.
The topic may be of interest to readers of journal. However, as it stands, the manuscript warrants substantial improvement before it can be considered for publication.
Based on the assumption that the article may be based on a large amount of work done by the authors, the issue of its publication in the journal could be further considered after its revision (for example, restructuring, annotation, discussion, etc.).

Round 2

Reviewer 2 Report

Dear authors! You have done a lot of work to correct my comments. The issue raised by the authors is indeed an important one, and is interesting from both scientific and practical perspectives. The structure of the article do meet the requirements of the publication (Research Manuscript Sections).
The introduction do contain a clear statement of the problem, the relevant literature on the subject, and a proposed approach or solution. The topicality and novelty of the research are understood from the introduction. The literature review refers to the central issue of the paper, it is quite extensive, relevant and thorough. The review will be of interest to other researchers. I would like to mention that the authors have comprehensively studied the literature on the issue published over the last five years. References are correct.
I believe that readers of this work will not be interested in reading conclusions and there is a discussion there. I would recommend dividing. Conclusions are one thing. This is the result of what you came to. And the discussion will lead to this conclusion.  But the authors see this, and this is your right!
However, I recommend this article for publication.
